Original research

# Addressing alcohol-related harms in the local night-time economy: a qualitative process evaluation from a complex systems perspective

Elizabeth McGill  , Dalya Marks, Mark Petticrew, Matt Egan

Department of Public Health, Environments and Society, London School of Hygiene & Tropical Medicine, London, UK

**Correspondence to**
Dr Elizabeth McGill;
elizabeth.mcgill@lshtm.ac.uk

## ABSTRACT

**Objectives** English local authorities (LAs) are interested in reducing alcohol-related harms and may use discretionary powers such as the Late Night Levy (LNL) to do so.
This study aims to describe how system stakeholders hypothesise the levy may generate changes and to explore how the system, its actors and the intervention adapt and co-evolve over time.

**Design** A process evaluation from a complex systems perspective, using qualitative methods.

**Setting** A London LA with high densities of residential and commercial properties, which implemented the LNL in 2014.

**Participants** Data were generated through interviews with LNL implementers and alcohol consumers, observations in bars and during LNL patrols and documentary review.

**Intervention** The LNL allows LAs to charge late-night alcohol retailers an annual fee (£299–£4440) to manage and police the night-time economy (NTE).

**Results** When the LNL was being considered, stakeholders from different interest groups advanced diverse opinions about its likely impacts while rarely referencing supporting research evidence. Proponents of the levy argued it could reduce crime and anti-social behaviour by providing additional funds to police and manage the NTE. Critics of the levy hypothesised adverse consequences linked to claims that the intervention would force venues to vary their hours or close, cluster closing times, reduce NTE diversity and undermine public–private partnerships. In the first 2 years, levy-funded patrols developed relationships with the licensed trade and the public. The LNL did not undermine public–private partnerships and while some premises varied their hours, these changes did not undermine the intervention's viability, nor significantly cluster venue closing times, nor obviously damage the area's reputation for having a diverse NTE.

**Conclusions** This study applies a framework for process evaluation from a complex systems perspective. The evaluation could be extended to measure alcohol-related outcomes and to consider the interplay between the national and local systems.

## STRENGTHS AND LIMITATIONS OF THIS STUDY

⇒ This study used a novel design, drawing on a complex systems perspective, to understand the mechanisms by which the intervention may generate system-wide changes.
⇒ We generated data through a range of qualitative methods, including interviews, observations and documentary review, which allowed us to collect data from a wide range of sources.
⇒ We include data from implementers, night-time economy users, business owners and staff but not health service workers.
⇒ The evaluation occurred after the intervention started, although many of the documents reviewed were produced prior to implementation.

## INTRODUCTION

In England, alcohol misuse is the largest risk factor for poor health and early mortality for adolescents and adults aged 15–49 years,[1] a pattern that is mirrored globally.[2] In addition to the health harms associated with alcohol consumption, alcohol contributes to broader societal harms including crime, violence, anti-social behaviour and disorder,[3] many of which occur within the context of the night-time economy (NTE).[4] Different approaches to reducing alcohol harms have been proposed and put into practice. For example, modifying alcohol availability (physical, temporal and economic),[5][6] policing and community safety interventions,[7][8] attempting to promote corporate responsibility,[9] encouraging certain types of alcohol outlets while discouraging others[10] and encouraging individuals to change consumption behaviours.[11] An evidence synthesis by Martineau *et al* found that evidence tended to support state-enforced regulations restricting the availability of alcohol for sale over non-regulatory approaches.[12] A recent critical review[6] of research on availability broadly reinforced this finding but noted limitations to the evidence base. For example, much of the evidence came from a narrow range of countries (particularly Australia and the USA) raising concerns about wider generalisablity and few studies examined effects of changing

temporal alcohol availability (hours of sale, including opening and closing hours). A number of studies from the UK and elsewhere have evaluated local-level regulation of alcohol availability and NTEs.[1 13–24] In England, changes to alcohol licensing systems that included new discretionary powers for local authorities (LAs) have been the focus of recent and ongoing research.[13–16 25]

At the turn of the 21st century in England, there was growing public discourse and concern about the rise of the 'alcohol-fuelled, consumption-driven, night-time high street' (Hadfield[26] p466), which was characterised by clusters of late-night establishments and a tension between those enjoying nights out, those employed within or profiting from the NTE and those impacted by violence, anti-social behaviour and nuisance.[27 28] A series of sweeping legislative and regulatory changes were made, with claims made that this would create a safer NTE, while generating economic benefits to businesses, the people they employed and to governments.[27 29] These changes included the Licensing Act 2003 (enacted in 2005), which transferred responsibility for alcohol licensing from magistrates to LAs and removed fixed closing times for alcohol-retailing venues.[29] The sale of alcohol in England is, therefore, overseen by LAs, also referred to as Councils, through licensing, trading standards and planning bodies.[30] In this context, LAs have access to a range of discretionary powers to tackle alcohol-related harms, including Cumulative Impact Policies and Early Morning Restriction Orders, both of which were introduced in the Licensing Act 2003.[31] Another discretionary power, which will be the focus of this evaluation, is the Late Night Levy (LNL), which was introduced in the 2011 Police Reform and Social Responsibility Act.

### Late Night Levy

The LNL was designed to 'empower local areas to charge businesses that supply alcohol late into the night for the extra enforcement costs that the NTE generates for police and licensing authorities' (Home Office[32] p1). The intervention aims to prevent and address disturbance and crime associated with late night drinking. The power is discretionary and LAs can choose, following a period of local consultation, to implement a levy on all establishments in the on-trade and off-trade that have a licence to sell alcohol between midnight and 6:00. The amount each premise pays is set out in a nationally determined fee schedule based on the rateable value of the premise and the degree to which the premise was primarily

alcohol led (table 1). Individual LAs may exempt certain types of premises, such as those operating within a Business Improvement District and/or offer reductions for premises engaging in schemes such as Best Bar None or PubWatch. BIDs, Best Bar None and PubWatch are business-led and alcohol industry-led schemes and businesses voluntarily participate in them. These initiatives are supported by public bodies, including LAs or the Home Office. As shorthand, these schemes will be referred to as public–private partnership (PPP) schemes. The revenue from the levy, following the deduction of administrative costs, must be split with a minimum of 70% going to the police and the remainder to the LA. In 2011, The Home Office estimated that the levy would likely be viable in 94 of the 378 LAs across England and Wales and generate a total net revenue of £12.1 m per year.[33] The legislation was enacted in 2011 and Newcastle City Council was the first to adopt the levy in November 2013.

### The LNL as an event in a complex system

Public health researchers have become increasingly interested in applying a complex systems perspective to analysing the multiple interactions that lead to patterns of health behaviour, outcomes and inequalities across communities.[34–36] Where LAs choose to implement the LNL, it is introduced locally into a complex system that interacts with regional, national and international systems. A system is a group of elements, bounded in some way, that interact with each other.[37 38] A complex system is one that is characterised by unpredictability and change over time.[39 40] Complex systems exhibit emergent properties that cannot be reduced to the behaviour of the individual system elements.[41] Elements within a system respond to internal and external system inputs; these responses may feedback on the inputs themselves, either amplifying or dampening their impacts, which may, in turn, create unanticipated or unintended effects.[42 43] Analysing a complex system encompasses making sense of the system's trajectory, considering how it is influenced by its previous history and the interactions between its elements.[40 44] Key concepts from a complex systems perspective, which we consider in this paper, are defined in table 2.

Complex systems are characterised by their open boundaries and as a result, they interact with, influence and are influenced by, other systems.[45] From a geographical perspective, they can be characterised by both horizontal and vertical complexities. Horizontal complexity refers to the relationships between system elements and

| Table 1 | Late Night Levy charges | | | | | | |
|---|---|---|---|---|---|---|---|
| **Rateable value** | **A: No rateable value – £4300** | **B: £4,301– £33000** | **C: £33,001– £87 000** | **D: £87,001– £125 000** | **E: £125,001+above** | **D x 2 multiplier applied to premises in category D that are primarily/exclusively alcohol-led** | **E x 3 multiplier applied to premises in category E that are primarily/exclusively alcohol-led** |
| Annual levy charge | £299 | £768 | £1259 | £1365 | £1493 | £2730 | £4440 |

Source: Home Office 2015.[82]

**Table 2** Complex system concepts

| Concept | Definition |
| --- | --- |
| Elements | Components within a system ('agents', institutions, resources, etc.)[40] |
| Boundaries | The 'limits' or 'bounds' of a given system; boundary judgements may be made by system actors (first-order) or researchers (second-order)[38 54] |
| Levels | The structure of the system; levels may operate horizontally and/or vertically depending on boundary decisions[43 83] |
| Relationships and interactions | Connections between different system elements, within and across system levels, and between elements and the broader context[84] |
| Local rules | The norms and principles that guide interactions between system elements and drive system behaviour[85] |
| Perspectives | The different ways actors within the system may view the system, their goals and actions and boundary decisions[86] |
| Non-linearity | Inputs into a system may lead to a non-correspondingly-sized impact[54] |
| Feedback | Responses that either amplify or dampen the impacts stemming from an intervention and may alter the intervention itself[42] |
| Adaptation | The ways in which system elements and the system as a whole respond in response to internal and external inputs[39] |
| Emergent properties | The emergent, collective behaviour of a system that cannot be reduced to its individual parts[87] |
| Co-evolution | The changes to a system and the broader systems in which it is located, over time[40] |
| Unintended consequences | Processes and impacts that were unanticipated at the design stage of an intervention[43] |
| System trajectories | The evolution of a system over time, which is path dependent or constrained in some ways due to its history[40 44] |

between systems within the same geographical scale. Vertical complexity refers to the relationships and interactions across geographical scales, with, for example, an emphasis on how international and national systems may influence, constrain and shape local systems.[46 47] A recent scoping review of complex systems' perspectives applied to alcohol consumption and prevention found that much of the research in this field is conducted in sub-local (eg, individual, families, social networks) or local (eg, neighbourhood, town, cities) systems. Far less consideration is given to the ways that the local systems interact with the national or international systems.[47]

A complex systems perspective applied to public health evaluation involves analysing the multiple ways in which a complex system and an intervention interact and influence each other to generate health impacts and health inequalities.[48–51] Evaluators might consider interventions as 'events' within systems that have the ability to disrupt system behaviour, generating evolving and adaptive patterns of behaviour and emergent outcomes.[44 52]

In public health, process evaluations have traditionally been used to understand the mechanisms by which interventions lead to impact, the influence of the broader context on observed variations in impact as well as to assess intervention fidelity and the quality of implementation.[53] Applying a complex systems' perspective to a process evaluation can be used to first describe the system, understand its elements, boundaries and the 'rules' or norms that govern the behaviour of its elements and the ways in which they interact each other. Following

the introduction of an intervention such as the LNL, a process evaluation with a complex systems perspective then aims to understand the mechanisms by which the elements within the system, and the system as a whole, adapt and co-evolve in response.

This process evaluation was conducted in one London LA with the dual aims of (1) describing the system into which the LNL is introduced and synthesising stakeholder hypotheses about the ways in which the levy may generate change within the system and (2) exploring how the intervention acts as an event within the system, with an emphasis on understanding how the system, its actors and the intervention adapt and co-evolve over time.

## METHODS
### Study design and data generation
We applied a framework for process evaluation using a complex systems perspective to data we collected on the LNL in one LA.[49] This evaluation framework consists of two phases: phase 1 involves producing a static system description and developing hypotheses of how the system may change in response to the intervention; phase 2 analyses the system as it undergoes change following implementation. The evaluation approach is adaptive and hypotheses generated in phase 1 are intended to inform the evaluative focus of phase 2. In phase 2, evaluators should be open to exploring unintended processes that stem from the intervention, which may not have

been considered at the design stage or in phase 1 of the evaluation.

The Standards for Reporting Qualitative Research checklist is provided in online supplemental material 1.

## Intervention and setting

The process evaluation was conducted in an English LA located in a metropolitan area with a large NTE. The LA held a consultation on the levy at the end of 2013 and implemented the LNL in late 2014. The levy hours are set at 00:01 to 06:00 and businesses that demonstrate commitment to best practice, as defined by the LA, are eligible for a 30% reduction of the levy fee. Businesses that are a member of the local BID, which requires members to pay a levy separate to the LNL, are neither exempt from the levy nor granted an automatic reduction in the fee. The Metropolitan Police and the LA chose to pool the net amount of levy payments to deliver one broad programme consisting of two different strands: (1) additional dedicated police resource to coordinate NTE policing and conduct support and enforcement activity and (2) a visible street-based patrol service delivered by a police-accredited community safety company four nights per week to give assistance to the licensed trade and members of the public. A LNL Board with representation from licensees oversees the use of funds raised through the levy.

## Sampling and data generation

A complex systems' perspective encourages evaluators to consider the intervention as a multi-stage process that, in the instance of the LNL, began with changes in national policy, then a local consultation and finally local implementation. Local delivery processes could continue to interact with national (or other 'non-local') developments. However, this evaluation focuses primarily on the local system: a focus on horizontal complexity. This local focus represents a 'secondary boundary judgement'[54]; that is, one that is made by evaluators (compared with a 'first-order boundary judgement', which is made by actors operating within the system).

The sampling strategy aimed to capture a range of different actors and perspectives within the national and local systems in order to contrast how different actors perceive, respond and adapt to the introduction of the intervention. Given the evaluative focus on the LNL in one LA, the sampling strategy was designed to primarily collect data from local actors through interviews, observations and a documentary analysis. However, recognising that complex systems are open systems, the sampling strategy was intentionally wider than the local system and the documentary analysis also included national data in order to analyse vertical systemic relationships.

In this process evaluation, phase 1 focuses on the period prior to local implementation, which included the national policy change and the local consultation. Phase 2 focuses on the local implementation stage and is the stage at which we became involved in evaluation.

Data collection for phase 1 was largely retrospective, but based on primary documentary sources generated during the earlier time period. Phase 2 was based on interviews, observations and document analysis collected during the first 2 years of the levy's implementation.

A range of data collection methods were used, including: a review of national and local documents, interviews with those implementing and delivering the LNL locally (n=12), interviews with users of the NTE (n=9), observations of community safety patrols (28.5 hours), which included informal conversations with patrol officers (n=10) and observations in pubs and bars (6 hours). Table 3 shows the documents analysed and their publication dates; Table 4 provides details of the primary data collection. To preserve participant anonymity, generic job roles are presented to remove identifying information. Data collection and fieldwork were conducted by EM, a research fellow with experience of a range of qualitative methods and analysis.

Documents were identified through online searches which included searches of national and local government websites for documents about the LNL, alcohol and health and crime and safety. In addition, Google searches were undertaken using the term 'late night levy'. Documents were included if they shed light on the rationale and process for developing and implementing the levy or reported on the levy following implementation. All documents are located in the public domain. Some of the documents included what might be considered 'outputs' in a process evaluation and short-term social and health impacts following intervention implementation. The analysis of these data focused on how they were presented, for what purposes, by which actors and how they suggested early indictors of change stemming from the intervention. We report some of the data from these documents in the Results section.

Interviews with professionals implementing and delivering the LA's (table 4), LNL followed a topic guide (online supplemental material 2) and asked participants about alcohol-related challenges, their experience of the LNL and the system in which the intervention is located. Topic guides were semi-structured to allow the participant scope to guide the conversation based on their experiences and understanding of the local system and the intervention. The interviews were audio-recorded and transcribed.

Observations were conducted during five community safety patrols partly funded by the LNL in which addressing alcohol consumption and associated harms was either a primary or secondary focus of the patrol. The observations were semi-structured; an observation template (online supplemental material 2) was used to systematically capture elements of the patrols as well as be open to capturing observations not envisaged at the research design stage. During the patrols, the fieldworker engaged in informal conversations with patrol staff and observed their actions and engagement with individuals or groups, including staff from licensed premises, police

**Table 3** Documents in documentary review

|  | Title | Organisation (Date) |
|---|---|---|
| National documents | Police Reform and Social Responsibility Bill Research Paper 10/81 | House of Commons Library (2010) |
|  | Impact Assessment for the Alcohol Measures in the Police Reform and Social Responsibility Bill | Home Office (2011) |
|  | Police Reform and Social Responsibility Act (2011) | Act of Parliament (2011) |
|  | The Government's Alcohol Strategy | Home Office (2012) |
|  | Next steps following the consultation on delivering the Government's Alcohol Strategy | Home Office (2013) |
|  | Amended guidance on the late night levy | Home Office (2015) |
|  | The late night levy | House of Commons Library (2015) |
|  | Modern Crime Prevention Strategy | Home Office (2016) |
|  | Policing and Crime Bill: Changes to the Late Night Levy—Impact Assessment | Home Office (2016) |
| Local documents | Annual Public Health Reports (n=5) | Council (2011–2016/17) |
|  | Licensing Policies (n=2) | Council (2011–2017) |
|  | LNL Consultation | Council (2013) |
|  | LNL Consultation Responses (n=338) | Council (2014) |
|  | LNL Written Consultation Responses (n=31) | Council (2014) |
|  | LNL Council Meeting Minutes | Licensing Committee (2014) |
|  | LNL Year 1 and Year 2 Reports | Council (2016; 2017) |
|  | LNL Year 1 and Year 2 Reports | Community Safety Company |
|  | BID Annual Reports | BID Board (2015/2016; 2016/2017) |

LNL, Late Night Levy.

officers, users of the NTE, street drinkers and rough sleepers. In total, 10 officers conducted the patrols, two of whom were also formally interviewed prior to the patrols. Throughout each patrol, the fieldworker wrote notes when appropriate and, where possible, captured direct quotations from patrol officers. An additional

**Table 4** Primary data collection

|  | Participants | Number (details) | Year |
|---|---|---|---|
| Interviews (n=21) (10.4 hours) | Local authority managers and officers relevant to licensing and public health | 4 (one individual interview; three interviewed as a group) | 2014 |
|  | Police officers | 3 (individual interviews) | 2016 |
|  | Community safety officers | 5 (two individual interviews; two interviewed as a pair) | 2014, 2016 |
|  | Users of the NTE | 9 (interviewed in pairs or one group of three; Fridays between 20:00 and 21:30) | 2016 |
| Observations and informal conversations (35.5 hours) | LNL-funded, community safety patrols; five different officers | 2 (18.5 hours; five officers; Friday 21:00–7:00 and Saturday 21:30–8:00) | 2016 |
|  | Non-levy, community safety patrols; five different officers | 3 (10 hours; five officers; Tuesday 6:00-9:00; Wednesday 13:00-20:00) | 2016 |
|  | Quarterly review meeting (local authority managers; community safety company managers) | 1 (1 hour; four participants) | 2016 |
|  | Pubs and bars (observation only) | 4 (6 hours; Fridays between 19:30 and 22:00) | 2016 |

LNL, Late Night Levy; NTE, night-time economy.

observation was conducted during an LNL review meeting between managers from the LA and the community safety company.

In order to better understand how users of the NTE experience the local alcohol system and the LNL, interviews were conducted in pubs and bars. Nine participants were recruited from alcohol-retailing venues; the fieldworker approached groups of 2–3 drinkers for semi-structured interviews following a topic guide (online supplemental material 2) that asked about the local area, particularly its NTE and their views on the LNL. Due to the setting, the fieldworker did not take notes during the interview or record the discussion. Notes, including any direct quotations, were written immediately following each interview.

## Patient and public involvement

No patients or public were involved.

## Analysis
### Phase 1

The framework for process evaluation from a complex systems perspective using qualitative methods suggests several questions to guide phase 1 of the evaluation: (1) What is the system of interest and what are its boundaries? (2) What are the characteristics of the system and how does it behave at the initial timepoint? (3) What are system stakeholders' perceptions about the ways in which the intervention could lead to changes within the system, including changes that may be unanticipated by intervention designers[49]? The 'Intervention and Setting' section above sets out the local system of interest and its boundaries, which for this evaluation, are the geographical boundaries of the LA. The third question is the focus of this phase of the evaluation.

The analysis began with an in-depth reading of all transcripts, fieldnotes and documents and a deductive approach to coding the data was undertaken, guided by a number of concepts from systems thinking which included elements, boundaries, levels, relationships and interactions, perspectives and history (see table 2). The coding process was used to make sense of the national and local histories that created the conditions for the development and implementation of the LNL, the goals of different actors and how their perspectives influenced their views towards the levy.

The emphasis of the analysis in phase 1 was to use the data to synthesise and articulate stakeholder hypotheses about the ways in which they believed the intervention could lead to changes within, and beyond, the system into which it is introduced. In order to do this, a coding framework in the form of a map was developed using Visual Understanding Environment software[55] (online supplemental material 3). In order to develop the framework, a list of variables relevant to the LNL, nationally and locally, were independently generated by two researchers (EM and ME) from the coded data. The variables and the relationships between them were then represented visually on a map in order to depict the ways in which stakeholders hypothesised the levy could generate change within the local system. The analysis of phase 1 was completed before the phase 2 analysis, so that it could inform the analytical focus for Phase 2.

### Phase 2

In phase 2 of the process evaluation framework, the evaluator seeks to understand how the system and the intervention itself change following implementation, exploring the mechanisms by which change occurs.[49] The hypotheses put forth by system stakeholders articulated in phase 1 were used to guide the analysis. In phase 2, the focus of the evaluation was on the new actors that were introduced into the system with levy funding. Guided by the four main hypotheses identified in phase 1, there was a simultaneous focus on the system elements and the system as a whole, considering how they adapt and co-evolve over time, disrupting the local system rules and patterns of behaviour. The coding and analysis were led by EM, with analytical discussions taking place across the research team. NVivo V.12 was used to aid the data analysis.[56]

## RESULTS
### Phase 1: system description and system stakeholders' hypotheses

System stakeholders articulated four hypotheses about the ways in which the levy could generate change within the local system. In the following section, each of these hypotheses will be described and visually represented.

#### Hypothesis 1: increased resources

The first hypothesis, as articulated by those designing, implementing and delivering the intervention, was that the LNL would increase the resources available to police and manage the NTE, which would be used for street-based community safety and policing and additional street cleaning services. These services, would, in turn, lead to a number of positive impacts for residents, visitors and commercial actors (figure 1):

> This will produce additional funding for the council and police to use to address the impacts and strains on local services that occur between midnight and 6am in [LA]. [...] we believe that the LNL can be used to reduce the instance of crime, disorder and anti-social behaviour during the levy hours as well as improve the local environment (LNL Consultation, 2013).

Some residents and visitors further described the mechanism by which such change would occur, placing an emphasis on the additional police and community safety presence; for example, one woman we interviewed in a pub believed more police on the street 'means I can walk home safely at 2:00' Others described police as a deterrent for anti-social behaviour and noise:

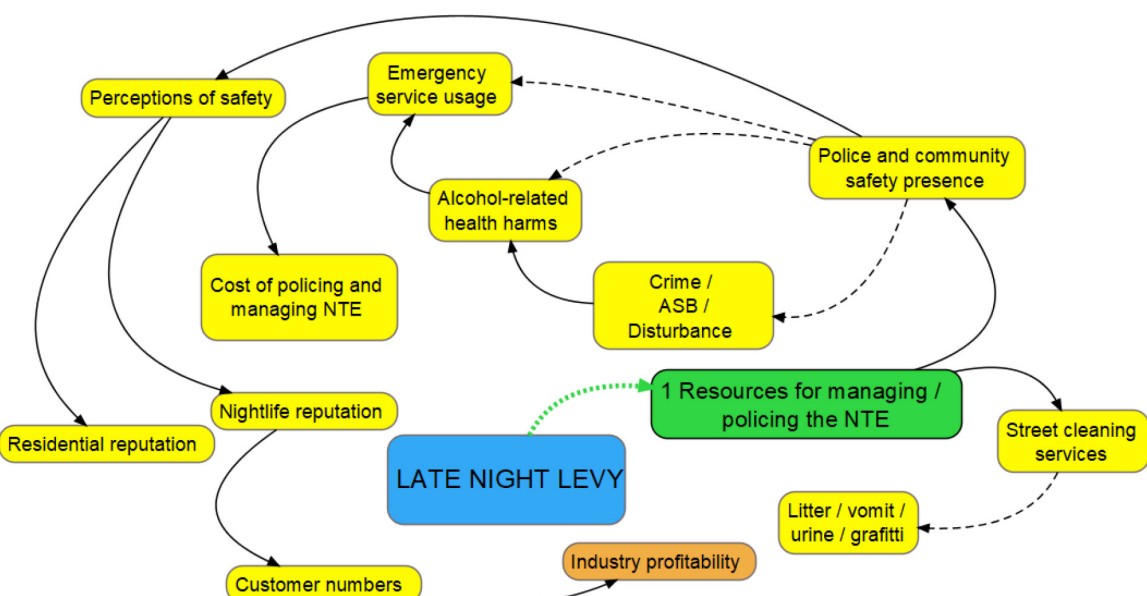

**Figure 1** Hypothesis 1. Orange bubble=national variable; yellow bubbles=local variables; green bubbles=immediate theorised impacts stemming from levy introduction. Solid line: positive relationship between variables; dashed line: inverse relationship between variables; dotted green lines: theorised impacts stemming from the levy introduction. ASB, anti-social behaviour; NTE, night-time economy.

I live on a side street of a late licensed premise and am woken up between 2 am and 7 am regularly every Saturday and Sunday morning. I don't think they realise the noise they're making so if there was a police presence I don't think they'd be as boisterous (consultation response, resident).

It was theorised that a safer NTE would enhance the overall reputation of the area, driving up visitor numbers and encouraging individuals to spend more money in local establishments. Implicit in this argument was that the levy fee would be easily offset by increased customer numbers, thereby making the levy beneficial for both commercial actors and the LA:

the money's going to pay for more policing, and [licensees] get more policing and that is beneficial for them because the safer an area is, the more people that come to the area and the more money that gets spent and the more money they make (interview, community safety officer).

### Hypothesis 2: reduced support for PPP schemes

A second hypothesis, articulated by businesses, was that licensees would be disinclined to continue to support PPP schemes because the resources do so would be redirected to paying the levy:

If operators do choose to pay the levy then it will impact on funding they can provide for partnership initiatives such as BIDs, Pubwatch and Best Bar None which the Council should look to support and promote in preference to a levy (consultation response, Pub company).

Many licensees, in particular, expressed concern about the LNL's impact on the BID, which funds dedicated police officers and cleaning services. In consultation responses, licensees agued they would vote against the BID when it came up for renewal, which in turn, would cause the BID to fail: (figure 2): 'As a BID payer if the levy were to come into force I would be voting no the next time the bid comes up for tender' (consultation response, licensee).

If the BID were to fail as a result of the LNL, licensees theorised that there would be a range of unintended consequences in the local system. These included reducing the overall resources available to manage and police the local area and damaging economic impacts because the BID is intended to create an environment that encourages residents and visitors to the area:

It is a possibility that nearly 40 licensed premises in the [local area] BID area will not vote for the BID again if this means that they pay two levies instead of only one. A BID needs a majority by numbers and also rateable value to succeed. A failure to achieve either one of these would, therefore, jeopardise the provision the BID makes for policing and cleaning as well as what we do to ensure a good shopping environment for local people, Christmas lights, hanging baskets, support for community events and much more (consultation response, Pub manager).

The local BID Board also perceived that any negative impact of the levy on the BID could generate impacts beyond the boundary of the local system. These system stakeholders argued that this hypothetical impact should be considered within the Mayor of London's goal to increase the number of BIDs throughout London:

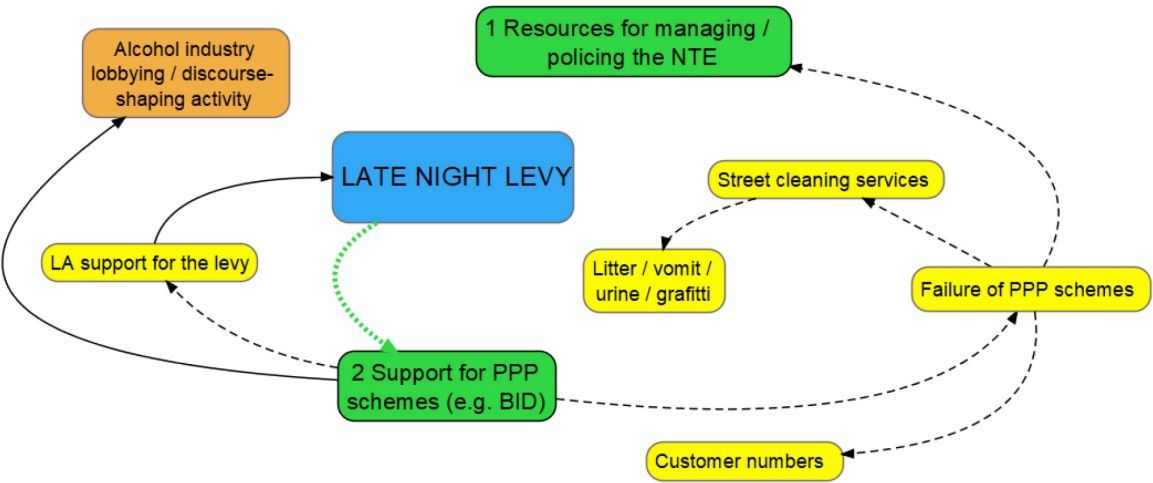

**Figure 2** Hypothesis 2. Orange bubble=national variable; yellow bubbles=local variables; green bubbles=immediate theorised impacts stemming from levy introduction. Solid line: positive relationship between variables; dashed line: inverse relationship between variables; dotted green lines: theorised impacts stemming from the levy introduction. LA, local authority; NTE, night-time economy; PPP, public–private partnership.

BIDs are burgeoning in London and the Mayor has set a target for a number of additional BIDs by 2015. It would be a loss, not just to [LA] but to London as a whole should [BID name] not get re-elected and become the first BID in London to fail (consultation response, BID Board).

### Hypotheses 3 and 4: premises will (3) vary hours or (4) close due to unwillingness or inability to pay the levy

In response to the Council's consultation, 42% of businesses reported they would voluntarily change their permitted licensing hours in response to the introduction of the levy. A smaller number argued that the levy would force some businesses to close as they became economically unviable. Consultation submissions from business hypothesised that these possible responses could lead to a range of unintended consequences, including undermining the levy, re-introducing a 'terminal hour,' reducing the diversity of late-night provision and ultimately generating negative economic consequences to the local area in the form of reduced employment and local investment (figure 3). Nearly all of these claims about potential impacts were made without reference to research evidence (and in fact, as will be raised in the Discussion, there is evidence that challenges these impact claims). However, one pub company's consultation submission did cite a study before going on to develop unsupported claims about the lack of a need for, and impact of, the levy. The study referenced[57 58] had found changes from fixed to staggered closing times were not associated with changes in overall violence but 'may have contributed to additional problems by spreading violence into the early hours of the morning' (Humphreys[58] p8). The pub company's submission referred to these study findings and then claimed they should be interpreted as meaning that more policing was not needed for the NTE and that the levy would not lead to extra policing:

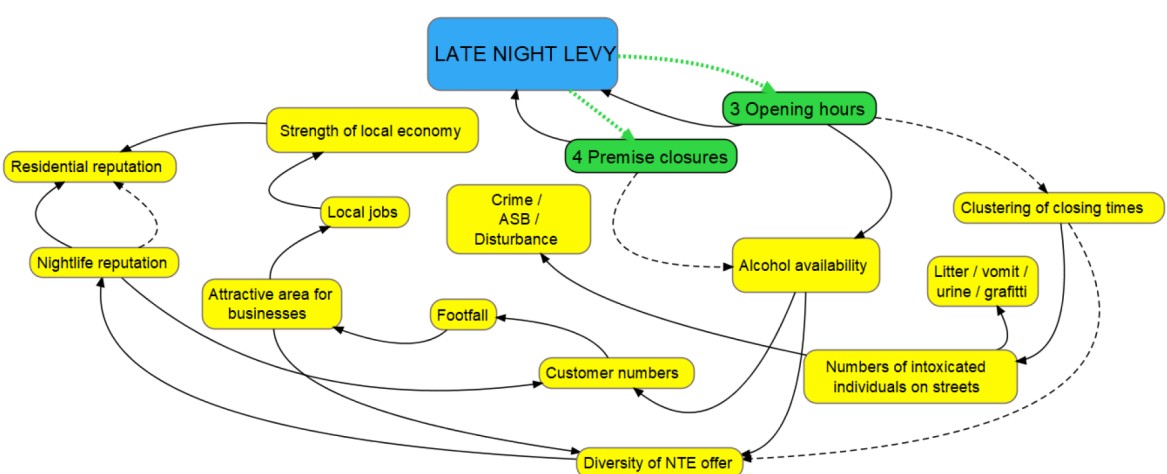

**Figure 3** Hypotheses 3 and 4. Orange bubble=national variable; yellow bubbles=local variables; green bubbles=immediate theorised impacts stemming from levy introduction. Solid line: positive relationship between variables; dashed line: inverse relationship between variables; dotted green lines: theorised impacts stemming from the levy introduction. ASB, anti-social behaviour; NTE, night-time economy.

Overall the net effect should be that same level of policing is required as previously. Cynics may suggest the Late Night Levy is a tax-raising measure to pay for the same level of policing that has always been provided (consultation response, Pub company).

Only two consultation responses, submitted by Public Health and the Clinical Commissioning Group, considered the LNL in terms of its ability to reduce alcohol consumption and associated harms by restricting the availability of alcohol. In contrast, all other system actors discussed the levy in terms of addressing the harms associated with acute intoxication, focusing primarily on disturbance, anti-social behaviour, crime and to a far lesser degree on health-related indicators such as ambulance call-outs or hospital admissions. In this sense, discourses around reducing or preventing alcohol consumption (primary prevention) were largely absent, with a focus instead on making the NTE a safer space for consumption and the possible economic and cultural impacts of the levy.

As businesses shut early, or closed entirely in response to the levy, business actors hypothesised that the LA would fail to generate sufficient revenue to provide the new proposed services: 'We remain to be convinced that the LNL will raise the amounts of money anticipated, as a significant number of permissions within (LA) are likely to be withdrawn, by way of the free minor variation procedure'. (consultation letter, Pub company and brewer). This represented an example of a perceived negative feedback loop; it was hypothesised that as fewer businesses remained to contribute to the levy through late-night provision, the ability for the levy to continue as an intervention would be jeopardised.

If businesses varied their operating hours to avoid the levy, some in the licensed trade argued, without reference to research evidence, that this could effectively reintroduce a 'terminal hour', whereby many premises close at the same time, which would lead to an increase in crime and anti-social behaviour:

If a number of premises reduce their hours as a result of the levy, this could potentially create anti-social behaviour issues with a large number of premises closing at the same time and a return to the spike of crime, disorder and nuisance and midnight observed across the country prior to the introduction of the Licensing Act 2003 (consultation response, trade organisation representing on-licence premises).

Some system actors expressed concern that smaller, independent businesses as well as those which are not alcohol-led, would be most affected by the levy, leaving a less diverse NTE dominated by pub chains and clubs. A reduction in diversity was theorised to make the LA less attractive, which, in turn, was hypothesised to have the potential of leading to negative economic impacts as customers choose to go elsewhere, moving beyond the boundaries of the LA:

Many operators will have to curtail their hours irrespective of the economic consequences, thereby reducing the number of post-midnight premises in the borough. […] visitors to the Borough's late night economy [would be] choosing other areas of London where no such restrictions apply with obvious economic consequences for [LA]'s late night economy and the businesses that rely on it (consultation response, operator of managed pubs).

In underscoring how elements of the system are interconnected, a number of businesses suggested that the LNL would have negative economic impacts that affect more than just late-night alcohol retailers, making the LA a less appealing area to operate a business:

I am currently looking at sites in the borough; I run a high end food and drink offer, if this levy is introduced I would have to look if the operation could still be viable. My venues do not run beyond midnight but I understand that early evening venues are intrinsically linked to the later venues and if these were to close or relocate it would reduce footfall in the areas affected (consultation response, prospective licensee).

Ultimately, many businesses argued that the LNL, through changes to opening hours, lower profit margins, premise closures and lack of investment in the local area, would result in negative economic consequences and job losses for the LA as a whole:

The council will further kill off the high street if they implement this levy. Pubs and bars will re-locate to other nearby locations where the levy is not in place and lose a number of job opportunities for local people. I thought the council's major objective as to increase employment opportunities for local people, not decrease it (consultation response).

### Phase 2: early implementation and mechanisms of system change
The levy began on 1 November 2014 and in the first year, fees were collected from 338 licence holders.

#### Hypothesis1: increased resources
The key hypothesis as described by those who designed and implemented the levy was that it would bring in additional resources to manage and police the NTE (figure 1). In the first year, the levy raised £397 278 and in the second year £377 122 (Council LNL Year 1 and 2 Reports). While these figures were lower than the Council's projected £450 000, the Council described these as sufficient to plug 'an identified gap' in managing and policing the NTE (Council, LNL Year 1 and 2 Reports). The additional resources were used to fund an NTE-specific police team and a four-person community safety patrol, delivered by a police-accredited, private company, that worked Thursday–Sunday nights from approximately 20:00–8:00. The new community safety service is the primary focus of phase 2 of this process evaluation; an

**Table 5** Community safety service

Patrol description: The patrol met at 20:00 and conducted a 'scan' of the borough, driving down main roads and stopping to address any issues they identified, such as visible pre-loading. At 22:00, the officers attended a briefing at the police station which included: (1) a police briefing for all officers on duty and (2) a NTE briefing for the NTE police patrol and the community safety officers. Following the briefing, the community safety officers patrolled the borough throughout the night, conducting a number of 'taskings' (which came from the Police, the Licensing Team or were self-generated), responding to calls from venues, identifying and responding to individuals and groups and patrolling areas where there were hyper-local 'kick-out times'. The patrol concluded around 8:00.

| Strands of the service | Year 1 | Year 2 |
|---|---|---|
| Welfare | 316 checks | 724 checks |
| Medical | 161 individuals | 97 individuals |
| Addressing anti-social behaviour, aggression, urination, pre-loading | 365 incidents of violent or aggressive behaviour, 451 dispersals, 738 warnings about conduct | 784 incidents of violent or aggressive behaviour, 675 dispersals, 1235 warnings about conduct |
| Support to the licensed trade | 2295 liaisons with licensed trade; 226 responses to calls | 2482 liaisons with licensed trade; 125 responses to calls |
| Intelligence gathering | 620 459 words | 620 292 words |

Source: Community Safety Company, LNL Year 1 and Year 2 Reports.
LNL, Late Night Levy; NTE, night-time economy.

overview of the structure of the new service is provided in table 5, along with output data from the service provider's annual reports.

A key component of the new patrol service, which significantly increased from year 1 to year 2, was engagement with users of the NTE to ensure their welfare and to intervene early in anti-social behaviour, disturbance and nuisance to prevent its escalation:

> So not only are they there to deal with the response side of it, but it's to try and prevent that happening in the first place, so to deal with those people who potentially would go on and commit further offences because they've started shouting and swearing and causing problems with someone up this end of the street. By the time you get down the other end, they've stopped in five other pubs on the other way, not been challenged, not been highlighted to anybody on the way down, although their behaviour's getting more and more rowdy. Then they go in, have a fight or cause a disturbance and need for police action further down the road (interview, police officer).

The welfare aspect of the service, which included community-safety officers helping members of the public, was also considered a critical component of the service and as shown in table 5, increased significantly from year 1 to 2. The officers also provided medical care and the medical service represented an evolution of the service. While it was always within their remit to have a first-aid trained officer, they expanded this provision shortly after starting the service and purchased additional medical equipment. In addition to supporting members of the public, the medical side of the service was seen as a low-cost mechanism to reduce the burden on the London Ambulance Service and NHS. In the first 2 years, the service reported preventing or cancelling the dispatch of 54 and 57 ambulances, respectively, which they calculated as savings of £16 200 and £14 478.

The Council reported a 17% reduction in alcohol-related crime between midnight and 8:00 and a 14.4% reduction in alcohol-associated violence compared with the previous 12 months, although they assumed that this was not all attributable to the levy. They also reported a large increase (29–30%) in calls to the police and anti-social behaviour line about alcohol-related incidents, which they further argued justified the need for the levy funding (Council, LNL Year 1 Report). In year 2, the council reported a 21% reduction in alcohol-related crimes compared with the previous 12 months, and a 24% decrease in anti-social behaviour calls (Council, LNL Year 2 Report).

### Engagement with the licensed trade

While hypothesis 1 emphasised the resources to police and manage NTE users in the area, the new patrol service also sought to develop relationships with local actors. Notably, they tried to develop relationships directly with the licensed trade—to monitor and support licensed operators to encourage safer business practices aimed at minimising anti-social behaviour within and outside the premises. In the first year of the levy funding, the patrol provided an introductory visit to 251 of the venues on the levy, which they argued was as an important mechanism to overcome hostility towards the levy and its funded patrols.

Outside of the initial visits, the patrol worked to develop relationships and trust with venue staff through the repeated interactions; a key element of this, which they contrasted with the police, was the deployment of the same officers every night, particularly in the first year of the service:

One of the things you absolutely have when you're any form of policing, really, you've got to have that consistency. You've got to have the relationships. That comes from, you know, repetition. It's from meeting the DPSs [designated premise supervisors] on a regular basis, building up a trust and an understanding of what you're there to do […]. Well if you're on rotation you can't possibly know. You wouldn't even know who that person is and you certainly wouldn't be able to kind of build a balanced intelligence picture (interview, community safety officer).

The Council in its report on the levy following the first year of implementation similarly underscored that the patrol was 'resourced by regular officers' and highlighted the relationships they developed with businesses:

Not only have the Nightsafe Patrol Officers have developed a good working relationship with licence holders and their door staff the team have acquired excellent working knowledge or the night-time economy in Islington and made a significant contribution to information gathered by the police and Local Authority (council, LNL Year 1 Report).

When probed, community safety and police officers described an evolution of the relationships such that many licensees began to engage with the service, overcoming their initial resistance:

We came up initially against a lot of unhappiness because it's another tax effectively, a levy on these premises. They don't want to pay it. They're already paying ridiculously high rates and other business taxes and stuff. So, but, you know, I get that. But we're seeing a change now, you know. A year, 18 months down the road, they can see a benefit to it, so […], if they need help they'll get help. You know, they'll prevent stuff happening and hopefully make their business more attractive (interview, police officer).

Others licensees, however, remained what the officers referred to as 'hostile venues,' continuing to oppose the levy and its associated services. Officers put this down to a misunderstanding of the service's remit: 'they (the licensees) see it as an enforcement role instead of a support role'. (excerpt from fieldnotes).

The community safety service was tasked with collecting intelligence to help the police or inform licensing decisions. Key to this intelligence-gather strategy was developing cooperative, rather than adversarial relationships with venue managers and staff, as described above. Information gathering and sharing among police, patrol and licensed venue operators was reciprocal, or in system terms, represented a positive feedback loop. Closer relationships among these three groups of actors appeared to emerge as a consequence, along with an 'othering' of certain venues who remained outside of this information sharing subsystem. Furthermore, the information gained was used to inform licensing decisions:

And [the community safety officers] assist us as well. Not just us as licensing officers, but the police on the whole, because within our briefings we can say to them, just little things that have happened, that you wouldn't normally get a chance to deal with, can you go and check on this and this, this, this, and just have a look and even in terms of where new applications are coming in and people are asking to do various different things in their licence, and we're thinking, not sure you could do that, but we need to check the place out. […]. And they report back to us, and then that assists us in saying whether someone can or can't have a licence. It's invaluable, really (interview, police licensing officer).

Through the mechanisms described above, the Council, the police and the community safety officers reported that more venues were operating in a 'responsible manner' following the implementation of the levy. Hence, while the initial hypothesis around extra resourcing focused on policing and management of NTE users, by the second year of LNL's implementation a new mechanism for impact had emerged through information sharing and relationship building between NTE operators and the agents that patrolled and policed the NTE:

Interviewer: do you think it (LNL and other licensing policies) has changed kind of how people consume alcohol in the borough?

Respondent (Police licensing officer): I don't think it's changed how people consume their alcohol in the borough. I think it's changed how operators operate.

Taken holistically, the new service was perceived to have changed how actors within the system behaved and interacted with one another, disrupting previous patterns of behaviour as system elements responded and adapted to the new services.

### Hypothesis 2: reduced support for PPP schemes

The second key hypothesis, articulated by businesses, was that if businesses were liable to pay the levy, they would no longer support PPP schemes, particularly the local BID (figure 2). This initial hypothesis did not accurately theorise how the system would adapt in the first 2 years of the intervention. Instead, in October 2016, members of the BID 'again voted resoundingly for us to continue' (BID website) and the BID expanded to cover a larger geographical area. Following the introduction of the levy, the BID reported a key priority for safety in their area was: 'achieving 24-hour security at (BID area) through co-ordinated working with street patrol (LNL-funded service)' (BID Annual Report, 2015/2016) and a licensee described a reliance on both BID-funded and LNL-funded patrols:

The night time economy is a major contributor to the wealth of the [BID area]. Making sure the environment is fun yet safe is a huge undertaking, not only for us licensees but also for the police and [LA] Council.

[BID name] makes sure we are all working together. Not only do we have the [BID-funded] Police Team at our disposal but can also rely on [LNL-funded service] (BID Annual Report, 2016/17, Bar Owner).

Prior to the levy's implementation, members of the licensed trade argued that the BID-funded services addressed their policing and safety needs. However, as the LNL-funded community safety patrol became embedded in the local system, some members of the BID came to see the community safety patrols as a complement to their own funded services and promoted collaboration between the two services, leading to greater resources for managing the local environment.

### Hypotheses 3 and 4: premises will (3) vary hours or (4) close due to unwillingness or inability to pay the levy

The final hypotheses were that a large number of premises would vary their hours in response to the introduction of the LNL, or in some cases close completely, which would lead to unanticipated consequences (figure 3). The data reported by the council showed that approximately one quarter of all premises who were initially liable to pay the levy either varied their licence hours or closed prior to the implementation date, which was lower than the 42% who indicated they would during the consultation period. The majority of these businesses varied their hours, rather than permanently closing their doors.

The majority of premises that were identified as being liable to pay the levy continued to operate after midnight and the LA did not see a re-introduction of a 'defacto terminal hour'. However, there remained clusters of bars and pubs that closed at similar time, which the community safety officers would refer to informally as 'kick-out times'.

Members of the licensed trade and some residents and visitors theorised the levy would create an NTE that lacked diversity, which in turn would drive down visitor numbers. During the course of fieldwork, we observed a busy NTE with bustling streets and busy venues. All the users of the NTE we spoke with during the course of fieldwork in the second year of the levy described numerous and diverse places to go out in the LA:

[Name] was talking about how there used to be only one place really to go (The Name—which she says is a great pub), but now there are so many options. The places to go out don't just include alcohol: "It used to be that there were just three places to eat … [she lists their names] and now there are so many to choose from (excerpt from fieldnotes).

Cumulatively, these data show that some premises did vary their hours in response to the introduction of the levy, but the levy remained viable and that an insufficient number of premises closed at midnight to reintroduce a 'terminal hour'. The LA maintained a reputation for providing a diverse and busy NTE following the implementation of the intervention.

## DISCUSSION AND CONCLUSION

This two-phased process evaluation sought to describe the local system into which the LNL is introduced and explore how the intervention may lead to changes as the system, its actors and the intervention adapt and co-evolve over time. We identified four main hypotheses put forward by system actors articulating the ways in which they envisaged the intervention would lead to system change, including those that were unanticipated at the intervention design stage.

In phase 1 of the evaluation, we analysed stakeholder opinions to develop the system map from which we would select hypotheses for analysis. During phase 1, we observed that stakeholders who were supportive of the LNL (and some stakeholders who were more critical) were pre-occupied with 'secondary prevention', which we define as policing and other services that aim to prevent intoxicated NTE customers engaging in violent, anti-social or risky behaviour. We contrast this with primary prevention, which aims to deter alcohol consumption in order to prevent intoxication that leads to social problems and to prevent harms to health caused by consumption.[59] A discourse that prioritises secondary prevention arguably aligns with commercial interests in that action to prevent harm is taken after the point of sale.

A second observation we make about our phase 1 analysis was that we found limited reference to research evidence in stakeholder consultation submissions. Submissions that cited research evidence directly were clear outliers. For those who promote evidence-informed decision-making, this is concerning. We do not assume that stakeholders were simply unwilling to explicitly cite their sources. Previous research on cultures of evidence among local practitioners has found a number of barriers to evidence use relevant to this study.[60] These include absence of evidence or inconclusive evidence, difficulty accessing evidence, prioritising local (often experiential) knowledge over published research of different interventions implemented in other contexts and a lack of interest in research that contradicts stakeholder opinions. A study of stakeholder submissions to an alcohol-related WHO consultation highlighted how evidence can be selected, misrepresented or ignored in ways that aligned with stakeholder interests.[61] In our study, the stakeholders who contributed to the LNL consultation could not be informed by an LNL evaluation, as none existed.

Nonetheless, there was still some research evidence available that could have informed stakeholder hypotheses about changes stemming from the levy. For example, some stakeholders claimed that the LNL would increase harms by clustering closing times of alcohol venues. However, this claim is weakened by evidence from an earlier Manchester (UK) study that found changes from fixed to staggered closing times were unrelated to changes in violence.[57 58] One stakeholder submission referred to this study but used its findings to make unsupported claims about the lack of need for more NTE policing and the lack of impact of the levy on policing. Since the

Manchester study, international evidence that NTE regulation can reduce harm has increased, most recently with an Australian study.[20–22] That study found that NTE regulation (that includes reduced trading hours) reduced alcohol harms without reducing the number of patrons, scale or diversity of NTEs and occurred while the number of alcohol licensed venues increased.[17 18 62]

This disconnect between stakeholder claims and research evidence raises an important point about the value of examining hypotheses derived from stakeholder opinions, when those opinions may lack supporting evidence or even be contradicted by evidence. We suggest a number of reasons why stakeholder-derived hypotheses should be examined. First, stakeholders either believe their claims or at least believe the claims serve some purpose (eg, self-interest), suggesting a continued need for scrutiny and, where appropriate, refutation. Second, it is possible that local stakeholders understand something about a particular context or intervention that is different from the contexts of previous research. Third, stakeholder opinions (even those contradicted by existing evidence) may have a powerful impact on (1) decisions to implement interventions, (2) willingness to comply with interventions and (3) lobbying efforts to discontinue interventions—all of which can be hypothesised to affect intervention impacts (eg, by blocking an intervention, making an intervention harder to implement or shortening an intervention's lifespan). We note that across England, only 10 LAs currently have an LNL in operation. In other areas, an LNL has been considered by an LA but abandoned at consultation stage or discontinued after implementation.[63 64] Our own findings also challenge stakeholder claims that the LNL would lead to unwanted consequences through clustering of closing times and reductions to NTE diversity and footfall. These findings contribute to a growing evidence base that, we hope, will encourage decision-makers to consider such claims with scepticism.

We turn now to each of the four hypotheses we examined in phase 2 of our study. The first was that the intervention would increase resources to police and manage the NTE. The intervention did help finance the introduction of new actors into the local system, who through consistent, visible and prolonged relationship building with the licensed trade and the public, sought to disrupt local system rules and develop new practices. Findings from the first 2 years of the levy suggest that these efforts led to an evolution in the way that many, although not all, licensees viewed the levy and a change in how some venues were managed.

Contrary to the expressed views of some stakeholders, the introduction of the LNL did not undermine PPP schemes during the study period, particularly the BID, as expressed in the second hypothesis. The LNL's implementation co-occurred with an increase in voluntary industry initiatives and partnerships. Previous research on PPPs and industry-led so-called 'social responsibility' activities have highlighted conflicts of interests and called into question their effectiveness in reducing or preventing harm.[1 65–72] The mechanisms by which increased regulation of the sale of harmful commodities might lead to increased voluntary and partnership activity warrants further investigation across different types of interventions and harmful commodities.

With regards to the third and fourth hypotheses, there was some evidence that premises varied their hours in response to the levy, but these changes did not ultimately undermine the viability of the levy, lead to the re-introduction of a terminal hour, nor obviously damage the NTE's reputation as being diverse and vibrant. Taken together, those in charge of developing and implementing the levy at the local level, viewed these early indications of system change as successful. This suggests a reinforcing feedback loop, whereby the perceived success of the levy in this LA ensured its continuation.

This evaluation represents the first application of our complex system framework for process evaluations.[49] While many in public health have argued that complex system approaches can produce better evidence for decision-making that account for real-world complexities,[36] there have been relatively few prominent examples of this perspective applied to public health process evaluation to date.[73] This work attempts to address some of these limitations. The use of the framework and explicit application of systems and complexity concepts was used to make sense of the broader system into which the levy is introduced, the many processes through which the levy may lead to impacts, many of which might be unanticipated and the dynamic responses to the intervention that lead to an evolution of the system's actors, their relationships with each other, the intervention and the system as a whole.

This process evaluation is also the first known evaluation of the LNL.[74 75] An Institute of Alcohol Studies (IAS) report reviewed the impact of the Licensing Act (2003) 10 years post-implementation and reported that the LNL had the potential to reduce alcohol availability by encouraging premises to shorten their opening hours, could help foster a cleaner environment through the provision of additional street cleaning resources and could be used to promote diversity in the NTE. The report also highlighted other possible impacts of the levy, including that the levy might prevent or damage partnership working between LAs and the alcohol industry, impact the industry's profitability, and be too inflexible a tool to be well suited to many LA's NTEs.[74] The findings from our process evaluation shed light on the mechanisms by which these impacts may or may not occur within a local system. Despite acknowledging that there has been no evaluation of the LNL's impact on crime and disorder, a subsequent joint IAS and Foundation for Alcohol Research and Education report argues: 'Attempts have been made to limit closing times in areas with acute problems, through the late night levy and early morning restriction order, although these policies have also proven largely ineffective' (Foster[75] p10). The IAS's judgement on the LNL is premature

in the absence of an impact evaluation that examines a range of health, social and economic outcomes. The Queensland Alcohol-related violence and Night-Time Economy (QUANTEM) project provides an example of how such a study could be conducted, using multiple data sources to measure impacts on alcohol-related health and social harms and local economies.[17–24] Furthermore, the ExILEnS study and research stemming from the NIHR School for Public Health Research alcohol programme demonstrate how impacts of local alcohol policies have been evaluated in UK contexts.[13 15 16 25] A well-theorised, robust impact evaluation of the LNL is overdue.

## Strengths and limitations

As evaluators, we made two crucial boundary decisions in this process evaluation: to focus on the local level and to include and exclude certain local system variables from our analysis. Together, these represent an emphasis on horizontal complexity. The first decision was made a priori and was influenced by the nature of the intervention (ie, a locally delivered intervention) and our interest in the delivery processes within one LA system. However, there are also vertical complexities that affect, influence and interact with the local system; the local system is embedded within broader regional, national and international systems and the boundaries between them are open.[38] We included some consideration of the national system in order to make sense of the context in which the LNL was introduced as a discretionary power available to LAs, but other stakeholders within the national and local systems or other evaluation teams might have chosen to broaden their boundaries. Furthermore, the focus on the local excludes learning and evidence from the international literature. Given limited evaluation resources and the adoption of a complex systems perspective, there was a trade-off of breadth versus depth. We chose to prioritise collecting data from a range of system stakeholders over a larger sample of any single stakeholder group. This choice was motivated by the aim of describing and analysing multiple perspectives and views held within the local system. A better resourced study could have included more depth as well as breadth.

We also made decisions about the variables of interest within the local system.[76] This was informed by the data generated through the evaluation and our aim was to focus on the variables we found to be most relevant to the LNL. Examples of this exist within the systems literature, for example, with researchers utilising data generated through documentary review and interviews to develop causal loop diagrams.[77] However, this raises important considerations around power dynamics and who ultimately decides where boundaries are drawn.[78] This work could fruitfully be extended by engaging in processes that invite system stakeholders to participate in the boundary decisions and critique[43 79] and to provide feedback on our synthesis of their perspectives. Specifically, our coding framework, which was depicted on a system map, would have benefitted from refinement by system stakeholders.

A limitation of this evaluation is that we did not collect primary data from residents or those working within the healthcare system. In addition, no data were generated or analysed about the broader economic impacts on the local economy. Conducting systems research often involves collecting data from a wide range of different actors across a given system,[39] which is resource-intensive and challenging when conducting smaller, local evaluations. Conducting a documentary analysis is one possible way to include data from a wider range of participants than might be possible through interviews and observations alone. In this evaluation, for example, we collected limited primary data from members of the licensed trade and relied on their extensive consultation responses, which provided insight into the ways in which they theorised that the levy might lead to a range of unanticipated impacts across the local system.

We collected data for phases 1 and 2 concurrently in the post-implementation period (although many documents included in our documentary review were produced prior to the levy's implementation). As a result, phase 1 informed the phase 2 analysis, but not the phase 2 data collection. While this approach underscores the flexibility of the process evaluation from a complex systems perspective framework, it also limited our ability to follow all emergent findings. For example, the consultation responses underscored how the levy might affect employment patterns in the local area, with premises having to vary their employees' shift patterns or make some employees redundant. We did not collect data from premises or from the LA that could then speak to these possible impacts.

## Future direction and conclusions

The process evaluation from a complex systems perspective is intended to be adaptive, drawing on early findings to inform subsequent data collection and analysis. A logical next step for this evaluative process is to measure alcohol-related outcomes and to understand the processes beyond the immediate local system of interest, to consider the vertical dimensions of complexity. The evaluation shed light on the possible spillover effects to neighbouring LAs, and these processes and outcomes could be explored. In addition, in the Modern Crime Prevention Strategy (2016), the Government proposed changing the structure of the levy to apply to specific types of premises or specific hotspots within the NTE, rather than entire LAs. At the end of the same year, the Home Office concluded that the levy had been implemented in fewer LAs that anticipated (n=7) because of criticisms 'that LAs consider the levy to be inflexible and the licensed trade has highlighted issues of unfairness in terms of which businesses pay the levy' (Home Office[80] p1) While the changes were to come into effect in 2020, at the time of writing, they have yet to do so. Finally, the COVID-19 pandemic represents a large system shock that has had significant financial impacts on LAs and the licensed trade.[81] An impact evaluation of the LNL is needed, accompanied by a further process evaluation

extended to explore these interacting local, regional, national and international processes and systems.

**Acknowledgements** We gratefully acknowledge discussions with Dr David Humphreys and Professor Niamh Fitzgerald that helped us during data interpretation.

**Contributors** EM designed the methodology, conducted data collection, led on the analysis and drafted this manuscript. ME contributed to the analysis. EM, DM, MP and ME provided input into data interpretation, critically revised the manuscript and approved the final version. EM is responsible for the overall content as guarantor.

**Funding** This study was funded by the National Institute for Health Research (NIHR) School for Public Health Research (SPHR), Grant Reference Number PD-SPH-2015. The views expressed are those of the authors and not necessarily those of the NIHR or the Department of Health and Social Care.

**Competing interests** None declared.

**Patient and public involvement** Patients and/or the public were not involved in the design, or conduct, or reporting, or dissemination plans of this research.

**Patient consent for publication** Not applicable.

**Ethics approval** This study involves human participants and was approved by The London School of Hygiene & Tropical Medicine Ethics Committee (ref: 10129). Participants gave informed consent to participate in the study before taking part.

**Provenance and peer review** Not commissioned; externally peer reviewed.

**Data availability statement** No data are available. No additional data are available. The ethical approval granted for this study stipulated that interview transcripts would not be shared beyond the research team.

**ORCID iD**
Elizabeth McGill http://orcid.org/0000-0002-3841-8467

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
