## [Reviewer comments · BMJ Open]

ARTICLE DETAILS

TITLE (PROVISIONAL)	Addressing alcohol-related harms in the local night-time economy: a qualitative process evaluation from a complex systems perspective
AUTHORS	McGill, Elizabeth; Marks, Dalya; Petticrew, Mark; Egan, Matt

VERSION 1 – REVIEW

REVIEWER	Miller, Peter Deakin University
REVIEW RETURNED	24-Jun-2021

GENERAL COMMENTS	thank you for the opportunity to review this paper. I think it describes a very important contextual response and policy example for the rest of the world. Overall, I think the article does a good job of describing this interesting and complicated intervention, though I of course have some suggestions for 'improvement'. Abstract: the abstract doesn't do a great job of giving your impression of the size or scope of a late-night levy. It would be good if you could give an example of the price range, simply two numbers with a hyphen between them in brackets. The second sentence in the results section of the abstract is very awkward and should be rewritten. One of the key issues in the results section and in the article more generally is that you replicate a bunch of myths perpetuated by the alcohol industry and police around the issue of clustering closing times and that earlier trading hours would somehow reduce diversity and produce negative impacts. The evidence is strongly counter this. You might like to see the new publications in Drug And Alcohol Review around the Queensland example, but there are many more examples of this. https://onlinelibrary.wiley.com/page/journal/14653362/homepage/special-issue-quantem Clustering closing times does not result in increased harm; very much the opposite because it usually pushes people home earlier. Similarly, the results from both Newcastle and Sydney in Australia have demonstrated substantial increases in diversity and economic benefits, despite the hype of media. You may wish to look at the repeated economic analyses conducted by Terry Bevan and Andrew license, via the capital city Lord Mayor's research: License, A., Edwards, A., & Bevan, T. (2018). Measuring the Australian Night Time Economy 2016-17. Retrieved from https://www.lordmayors.org/?p=1328
---

	it's important that this article and your work more generally doesn't replicate such myths. Introduction You really need to move the description of the late-night levy up into the introduction and fairly early on, I would say before the third paragraph. Otherwise I end up getting annoyed that I still don't really understand the details of it before you launch into a justification of complex systems- which I find less relevant. Methods It would be really helpful if a list of all documents, but those are included and those that were not, would be supplied in supplementary material. I understand that this is a bit more work, but in the end, it's important to be transparent about what you included and what you did not. Having ready access to these documents would help the reader. Otherwise, there is no real way for us to determine whether you reliably sampled everything; the equivalent of a response rate or random sampling. Description of "interviews with professionals implementing..." Should contain a link back to table 2. The observations conducted during safety patrols represent a bit of a missed opportunity to me. I think it would have been good if there was key data being collected systematically in the moment regarding the nature of the safety patrols, how well they interacted with patrons, levels of engagement with stakeholders, et cetera. A range of key outcomes could have been easily collected and would have been really helpful in thinking systematically about what the safety patrols did and did not do. I'm not clear on the value of presenting any sort of data from the fieldworker approaching groups of 2 to 3 drinkers but not keeping systematic notes. To be honest, this seems like poor practice and should not be encouraged. It certainly shouldn't inform any sort of interpretation, given the huge potential first subjective bias. I think this should be removed from the paper and am surprised that past any sort of ethical/methodological review. It certainly doesn't fill me with confidence regarding the notion of a complex system perspective. It suggests a very real opportunity for bias to creep in and a number of different data levels where systematic data collection has been done before in many studies and should be considered best practice. The fact that subsequent deductive reasoning was then undertaken again makes the data and process translucent and leaves me with way too many questions about what was included and why. "Ethical approval for this study was obtained from the s with particular attention paid to consent and safety issues around collecting data in situations where alcohol is consumed." - This needs to be corrected "A list of variables relevant to the LNL, nationally and locally, was independently generated by two researchers (EM and ME) from the coded data." - This seems to be somewhat open to bias. Only looking nationally and locally means that lessons from overseas could be ignored. It would seem that this would be a good opportunity to consult with key stakeholders or at least the research team and some sort of advisory committee. Figure 1.
--	--

	This diagram is overly complex and not only replicates myths, but has superfluous information as well as misses other factors. It needs to be much more refined. Three examples, but there are many more: Linking clustering of closing times to diversity of night life Having separate footfall/customer numbers Linking number of intox people on street to closing time cluster I sincerely doubt the support for a PPP is related to its impact; in fact, one might suggest that they are popular because they are ineffective. There's a natural progression through the night and most people go home by around 2-3am. While there used to be an issue of pushing people out at 11, even extending to 1 am means many people will be starting to head home naturally. I'm also left with the feeling that this paper tries to do too much and tries to address too many issues with not terribly reliable data. An example is the narratives on engagement with the licensed trade "One of the things you absolutely have when you're any form of policing, really, you've got to have that consistency. You've got to have the relationships. That comes from, you know, repetition. It's from meeting the DPSs [designated premise supervisors] on a regular basis, building up a trust and an understanding of what you're there to do [...]. Well if you're on rotation you can't possibly know. You wouldn't even know who that person is and you certainly wouldn't be able to kind of build a balanced intelligence picture. (Interview, Community safety officer)" it just has the feeling of uninformed anecdote. Is it really the case? There's no other data to talk about this and it's presented unproblematically. It's just as easy to draw from the section that nicer venues engage, not so good ones don't. That's what we've found with liquor accords in Australia. Discussion I find there's a lack of engagement with the research evidence in this space. There's large swathes of discussion without reference to other research. The strengths and limitations sections was comprehensive.
--	--

VERSION 1 – AUTHOR RESPONSE

Reviewer: 1

Prof. Peter Miller, Deakin University

Comments to the Author:

1. Thank you for the opportunity to review this paper. I think it describes a very important contextual response and policy example for the rest of the world.

Overall, I think the article does a good job of describing this interesting and complicated intervention, though I of course have some suggestions for 'improvement'.

Author response: Thank you for taking the time to review our manuscript and for suggesting ways in which to improve it.

Abstract:

2. The abstract doesn't do a great job of giving your impression of the size or scope of a late-night levy. It would be good if you could give an example of the price range, simply two numbers with a hyphen between them in brackets.

Author response: We have amended the abstract so that the range of the levy fee is now included. The sentence about the intervention now reads: "The LNL allows LAs to charge late-night alcohol retailers an annual fee (£299 - £4,440) to manage and police the night-time economy (NTE)."

3. The second sentence in the results section of the abstract is very awkward and should be rewritten.

Author response: We have re-written the first half of the results section of the abstract in response to this comment and the one below it. The text now reads: "When the LNL was being considered, stakeholders from different interest groups advanced diverse opinions about its likely impacts without referencing supporting research evidence. Proponents of the levy argued it could reduce crime and anti-social behaviour by providing additional funds to police and manage the NTE. Critics of the levy hypothesised adverse consequences linked to claims that the intervention would force venues to vary their hours or close, cluster closing times, reduce NTE diversity, and undermine public-private partnerships."

4. One of the key issues in the results section and in the article more generally is that you replicate a bunch of myths perpetuated by the alcohol industry and police around the issue of clustering closing times and that earlier trading hours would somehow reduce diversity and produce negative impacts. The evidence is strongly counter this. You might like to see the new publications in Drug And Alcohol Review around the Queensland example, but there are many more examples of this.

<https://onlinelibrary.wiley.com/page/journal/14653362/homepage/special-issue-quantem>

Clustering closing times does not result in increased harm; very much the opposite because it usually pushes people home earlier.

Similarly, the results from both Newcastle and Sydney in Australia have demonstrated substantial increases in diversity and economic benefits, despite the hype of media. You may wish to look at the repeated economic analyses conducted by Terry Bevan and Andrew License, via the capital city Lord Mayor's research:

License, A., Edwards, A., & Bevan, T. (2018). Measuring the Australian Night Time Economy 2016-17. Retrieved from <https://www.lordmayors.org/?p=1328>

it's important that this article and your work more generally doesn't replicate such myths.

Author response: The results section of the abstract has been re-written to make clearer that these were opinions of stakeholders advanced at the time point when the LNL was being considered by the local authority, and that opinions advanced (for example, during the formal consultation

process) were not based – or at least did not attempt to reference - research evidence. See the change of text presented in our response to Reviewer point 3 for the abstract.

In addition, we have made more clear throughout the manuscript that the Phase 1 data are the perceptions and views of stakeholders within the system about the hypothetical impacts of the levy. This is now discussed at some length in our revised Discussion. We have also included a more detailed discussion of published research in the Discussion section. We come back to this again in some of our responses below.

Thank you for highlighting some of the literature. You have prompted us to draw more attention to the way discussions about this intervention were conducted by stakeholders without attempting to consider research evidence – it is obviously a finding in its own right, particularly as these stakeholder opinions (whether they are supported by, contradicted by, or made in the absence of robust evidence) have an important role in shaping the acceptability, feasibility and implementation of the LNL. We come back to this below in response to some of the subsequent comments. We agree that when we discuss perceptions and arguments put forward by stakeholders about the intervention, it is important that we inform the reader of contradictory evidence and avoid replicating myths by giving the impression that we (the researchers) endorse them.

Introduction

5. You really need to move the description of the late-night levy up into the introduction and fairly early on, I would say before the third paragraph. Otherwise I end up getting annoyed that I still don't really understand the details of it before you launch into a justification of complex systems-which I find less relevant.

Author response: As suggested, we have now moved the description of the intervention to the Introduction before the third paragraph. We agree, it reads better this way.

Methods

6. It would be really helpful if a list of all documents, but those are included and those that were not, would be supplied in supplementary material. I understand that this is a bit more work, but in the end, it's important to be transparent about what you included and what you did not. Having ready access to these documents would help the reader. Otherwise, there is no real way for us to determine whether you reliably sampled everything; the equivalent of a response rate or random sampling.

Author response: Table 3 provides a list of all the documents that were included in the analysis. The table we moved to the new Supplementary Material 3 document (Table 1 in that document) lists each variable included in our system map, with a description and an example from our documentary analysis or primary data collection. That table references the quoted data source so we consider this to be transparent: i.e. the source of example is cited in the table, which allows the reader to see from where we are drawing our findings. While the majority of the documents relevant to Phase 1 of the evaluation are referenced in this table, we note that qualitative research tends not to refer directly to every single evidence source examined. By this, we mean that just as a qualitative research article reporting on (for example) semi-structured participant interviews does not necessarily quote from every participant who was interviewed, so an article reporting on a qualitative analysis of documents may not necessarily quote from every document examined. Our aim was to select, organise and consider a diverse range of views – both in our analysis of documents and our participants interviews.

7. Description of "interviews with professionals implementing..." Should contain a link back to table 2.

Author response: We have referred the reader back to Table 4 where we tabulate our primary data collection sample.

8. *The observations conducted during safety patrols represent a bit of a missed opportunity to me. I think it would have been good if there was key data being collected systematically in the moment regarding the nature of the safety patrols, how well they interacted with patrons, levels of engagement with stakeholders, et cetera. A range of key outcomes could have been easily collected and would have been really helpful in thinking systematically about what the safety patrols did and did not do.*

Author response: In response to this comment, we have now created a supplementary file (Supplementary Material 2) which provides the templates and topic guides that guided our observations and interviews. These tools were designed to be semi-structured so that the fieldworker was guided to systematically capture observations, as well as be open to capturing additional data not envisaged at the research design stage. We note that qualitative research data collection can take a variety of forms, ranging from unstructured to different degrees of structuring. For example, some qualitative research uses semi-structured interviews in controlled settings and more ethnographic / observational approaches in uncontrolled field-settings, and even unstructured participant and non-participant observation. We do not consider any of these approaches to be invalid, although in our case we opted for a mixture of interviews in controlled settings and semi-structured observations and interviews in uncontrolled settings. The former often has the advantage of providing researcher with time to explore topics in depth and better conditions for recording data; the latter has the advantage of giving the researcher/observer more direct access to the contexts under investigation and the people interacting directly in those contexts.

9. *I'm not clear on the value of presenting any sort of data from the fieldworker approaching groups of 2 to 3 drinkers but not keeping systematic notes. To be honest, this seems like poor practice and should not be encouraged. It certainly shouldn't inform any sort of interpretation, given the huge potential first subjective bias. I think this should be removed from the paper and am surprised that past any sort of ethical/methodological review.*

It certainly doesn't fill me with confidence regarding the notion of a complex system perspective. It suggests a very real opportunity for bias to creep in and a number of different data levels where systematic data collection has been done before in many studies and should be considered best practice. The fact that subsequent deductive reasoning was then undertaken again makes the data and process translucent and leaves me with way too many questions about what was included and why.

Author response: Drinkers were approached for an interview in groups of 2-3 and the semi-structured interviews were informed by a topic guide. In response to reviewer comments, we now present the topic guide in the new Supplementary Material 2 document. Notes were taken immediately following each interview. They were systematically taken in the sense that they were (semi) structured according to the items included in the topic guide. Given the noisy environments in which these interviews were conducted, it was not possible to conduct any audio recording. The protocol underwent a rigorous ethical approval process by the London School of Hygiene & Tropical Medicine's Ethics Committee.

10. *"Ethical approval for this study was obtained from the s with particular attention paid to consent and safety issues around collecting data in situations where alcohol is consumed."*

- *This needs to be corrected*

Author response: This sentence now reads: "Ethical approval for this study was obtained from the London School of Hygiene & Tropical Medicine Ethics Committee (ref: 10129) with particular attention paid to consent and safety issues around collecting data in situations where alcohol is consumed."

11. *"A list of variables relevant to the LNL, nationally and locally, was independently generated by two researchers (EM and ME) from the coded data."*

- *This seems to be somewhat open to bias. Only looking nationally and locally means that lessons from overseas could be ignored. It would seem that this would be a good opportunity to consult with key stakeholders or at least the research team and some sort of advisory committee.*

Author response: Our focus on the local stakeholder opinions for developing our Phase 1 hypotheses meant that we made their perceptions the priority. Those perceptions were typically local in focus and made virtually no reference to research evidence from any jurisdiction (UK or international). Nonetheless, we agree with the Reviewer that we should have made more reference to research evidence in the Introduction and Discussion section, which we have now done – see our response to point 14.

Our map and the included variables were utilised as a coding framework. In this process, two researchers agreed the framework. This comment raises an important point that it is good practice to put the framework to stakeholders for a refinement stage. We have now noted this as a limitation in our Discussion section. We did reach out to the local authority regarding this research to ask if they would be interested in hearing our findings and providing feedback. We did not receive a reply and which may well have been as a result of Covid-related workload pressures (we and colleagues were generally finding it hard to engage with LA practitioners across a number of projects during the height of the pandemic and lockdown – even LAs that we had generally good connections with, such as this one).

We have made sure to emphasise throughout the manuscript that this work represents a synthesis of stakeholder perspectives, rather than a definite view of the system of interest. This study is an application of systems thinking to qualitative methods where the focus is on subjective views and perceptions of different stakeholders. This contrasts with, for example, a system designed to be used for computational modelling where the aim is often to obtain estimates of effect sizes for particular outcomes of interest in the system. Please also see our response to the next point.

12. *Figure 1.*

This diagram is overly complex and not only replicates myths, but has superfluous information as well as misses other factors. It needs to be much more refined.

Three examples, but there are many more:

Linking clustering of closing times to diversity of night life

Having separate footfall/customer numbers

Linking number of intoxicated people on street to closing time cluster

I sincerely doubt the support for a PPP is related to its impact; in fact, one might suggest that they are popular because they are ineffective.

There's a natural progression through the night and most people go home by around 2-3am. While there used to be an issue of pushing people out at 11, even extending to 1 am means many people will be starting to head home naturally.

Author response: The map is a synthesis of stakeholder perceptions and is used to generate hypotheses that the second stage of this qualitative study then explores in more detail. To make this point more explicit, we have changed the language around 'theories of change' to 'hypotheses' in order to underscore that each of these reflect the ways in which stakeholders embedded within the local system hypothesised the levy to lead to a range of different consequences. As we now discuss in the article, these hypotheses rarely referenced research evidence and, indeed, are at times contradicted by published evidence. The reviewer raises a hugely important point – why would we want to explore hypotheses that includes stakeholder claims that not supported by research evidence? There are a number of reasons. Firstly, stakeholders either believe the points they are making, or at least think it serves their interests to make those points. Such claims continue to require examination and, where appropriate, refutation. Secondly, it is possible that local stakeholders understand something about their particular context and this particular intervention that is different

from the interventions and contexts that form the basis of other research. Thirdly, these stakeholder opinions (even those refuted by existing evidence) can have a powerful impact on (i) decisions to implement an intervention, (ii) peoples willingness to comply with interventions, and (iii) lobbying efforts to discontinue existing interventions – all of which can be hypothesised to affect the intervention’s impacts (e.g. by preventing an intervention, or making an intervention harder to implement effectively, or by shortening an intervention’s lifespan). We note that only 11 LAs have implemented a LNL: in other cases it has been considered by LAs but abandoned at (or before) consultation stage or discontinued after implementation. As reported in this paper, our exploration of claims around how the LNL would affect clustering, footfall and diversity suggests such claims were unfounded – and we hope this finding will encourage decision makers to reconsider arguments of this kind made against LNLs with a more critical eye (see p.23 of the revised manuscript). Nonetheless, we reiterate what we say in responses to point 11 and 14 – we definitely accept the Reviewer’s point that the article should include a more detailed discussion about what previously published research evidence does tell us. We have revised the Discussion section, notably with the addition of a substantial amount of new text at the start discussing the points raised here. We have decided to now move the system map to a supplementary material file as it forms our coding framework and is part of our analytical process, but we think it is likely overly complicated to be particularly meaningful to readers. We do think it is important that we continue to present this framework in order to be transparent about our analytical process; the map is a synthesis of stakeholder perception and was an initial analytical step for us before focussing in on the four key hypotheses we identified. We believe we should not make post-hoc adjustments to the map it at this stage because it informed our analysis.

13. I’m also left with the feeling that this paper tries to do too much and tries to address too many issues with not terribly reliable data.

An example is the narratives on engagement with the licensed trade “One of the things you absolutely have when you’re any form of policing, really, you’ve got to have that consistency. You’ve got to have the relationships. That comes from, you know, repetition. It’s from meeting the DPSs [designated premise supervisors] on a regular basis, building up a trust and an understanding of what you’re there to do [...]. Well if you’re on rotation you can’t possibly know. You wouldn’t even know who that person is and you certainly wouldn’t be able to kind of build a balanced intelligence picture. (Interview, Community safety officer)”

it just has the feeling of uninformed anecdote. Is it really the case? There’s no other data to talk about this and it’s presented unproblematically. It’s just as easy to draw from the section that nicer venues engage, not so good ones don’t. That’s what we’ve found with liquor accords in Australia.

Author response: In order to address the comment that the paper is trying to do too much, we have tightened the aim of the paper to focus more specifically on the identification and exploration of the four key hypotheses put forth by system stakeholders about the ways in which the levy could generate change within the system. We have therefore removed the part about ‘visually depicting’ the local system. We believe doing so tightens the scope of the paper. We now focus more explicitly on stakeholder perceptions and hypotheses, underscoring that these are generally not rooted in research evidence.

With regards to the comment about the narrative on engagement with the licensed trade, we have included an additional excerpt of data to show that this viewpoint was put forth by other system stakeholders as well:

“The Council in its report on the levy following the first year of implementation similarly underscored that the patrol was “resourced by regular officers” and highlighted the relationships they developed with businesses:

Not only have the Nightsafe Patrol Officers have developed a good working relationship with licence holders and their door staff the team have acquired excellent working knowledge of the night-time economy in Islington and made a significant contribution to information gathered by the police and Local Authority. (Council, LNL Year 1 Report)”

We would also say that the binary distinction between nice venues that engage and not nice venues that do not engage is recognisable in our own study setting – but it is still a heuristic. There are venues that exhibit both some nice/engaged and some less nice/non-engaged characteristics and some potential for fluidity over time. There’s also a sense that resistance to interventions like this can sometimes peak and trough – for example, peaking at planning and consultation stage when stakeholders perceive some opportunity to prevent the intervention occurring, and then in some quarters ebbing during the implementation period, as at least some stakeholders decide it is now in their interest to work with the new situation/system.

Discussion

14. I find there’s a lack of engagement with the research evidence in this space. There’s large swathes of discussion without reference to other research.

Author response:

We have now added more references to both the Introduction (first paragraph) and Discussion (throughout and notably the third and fourth paragraph and paragraph prior to the Strengths and Limitations section). In total, we have included an additional 33 references. Thank you for directing our attention to the new research from Australia – the dates of publication suggest that they were published after we submitted the original manuscript but it is great to be able to refer to them now (and of course there are earlier publications that could have been cited in the original manuscript).

15. The strengths and limitations sections was comprehensive.

Author response: Thank you for this comment, although in response to previous comments we have added some additional limitations to this section.

Reviewer: 1

Competing interests of Reviewer: none

VERSION 2 – REVIEW

REVIEWER	Miller, Peter Deakin University
REVIEW RETURNED	24-May-2022
GENERAL COMMENTS	The authors have done an excellent job of responding to my concerns and I believe have surpassed my requests and shown exemplary scholarship. I look forward to seeing the article published.